# Independent evolution of functionally exchangeable mitochondrial outer membrane import complexes

**Daniela G Vitali[1†], Sandro Käser[2†], Antonia Kolb[1†], Kai S Dimmer[1], Andre Schneider[2]\*, Doron Rapaport[1]\***

[1]Interfaculty Institute of Biochemistry, University of Tübingen, Tübingen, Germany; [2]Department of Chemistry and Biochemistry, University of Bern, Bern, Switzerland

**Abstract** Assembly and/or insertion of a subset of mitochondrial outer membrane (MOM) proteins, including subunits of the main MOM translocase, require the fungi-specific Mim1/Mim2 complex. So far it was unclear which proteins accomplish this task in other eukaryotes. Here, we show by reciprocal complementation that the MOM protein pATOM36 of trypanosomes is a functional analogue of yeast Mim1/Mim2 complex, even though these proteins show neither sequence nor topological similarity. Expression of pATOM36 rescues almost all growth, mitochondrial biogenesis, and morphology defects in yeast cells lacking Mim1 and/or Mim2. Conversely, co-expression of Mim1 and Mim2 restores the assembly and/or insertion defects of MOM proteins in trypanosomes ablated for pATOM36. Mim1/Mim2 and pATOM36 form native-like complexes when heterologously expressed, indicating that additional proteins are not part of these structures. Our findings indicate that Mim1/Mim2 and pATOM36 are the products of convergent evolution and arose only after the ancestors of fungi and trypanosomatids diverged.
DOI: https://doi.org/10.7554/eLife.34488.001

\*For correspondence:
andre.schneider@dcb.unibe.ch
(AS);
doron.rapaport@uni-tuebingen.de
(DR)

†These authors contributed equally to this work

**Competing interests:** The authors declare that no competing interests exist.

## Introduction

Mitochondrial outer membrane (MOM) proteins include a diverse set of enzymes, components of protein import machineries, pore forming proteins, as well as proteins mediating mitochondrial fusion, fission, and motility. In addition, the MOM harbours proteins that regulate apoptosis and mitophagy and hence are of central importance for the fate of the organelle and the whole cell. All these MOM proteins are nuclear-encoded and synthesised on cytosolic ribosomes. Therefore, they have to bear appropriate signals that ensure both their correct targeting to the organelle and their ability to acquire different topologies in the lipid bilayer. Despite their well-recognised importance, the diverse molecular mechanisms by which MOM proteins are specifically targeted to the organelle and inserted into their target membrane remain incompletely defined (*Dukanovic and Rapaport, 2011*).

MOM proteins can be divided into several topological groups (*Dukanovic and Rapaport, 2011*). Some of them span the lipid bilayer with one transmembrane segment (TMS), while others transverse the membrane with multiple β-strands or α-helical structures. Depending on their orientation, single-span proteins can be classified into three groups: the first two are signal- or tail-anchored proteins, which face the intermembrane space (IMS) with either the N- or C-terminus, respectively. These proteins typically expose the bulk of the protein to the cytosol and only a very short segment faces the IMS. A third subclass of single-span proteins exposes soluble domains towards both the IMS and the cytosol. Other integral MOM proteins span the bilayer either with several α-helical TMSs or as β-barrel structures. Whereas the import pathway taken by β-barrel precursor proteins has been studied in some detail (*Becker et al., 2008b*; *Endo and Yamano, 2009*; *Walther et al.,*

*2009*), much less is known about the factors and the mechanisms that assure the membrane integration of MOM proteins with helical TMSs.

MOM helical multispan proteins follow a unique import pathway in yeast cells (*Becker et al., 2011*; *Papic et al., 2011*). Precursors of these proteins are integrated into the membrane in a process where the MOM protein *m*itochondrial *im*port 1 (Mim1) cooperates with the import receptor Tom70 in binding precursor proteins and facilitating their insertion into the lipid bilayer. Interestingly, it appears that neither other subunits of the translocase of the outer membrane (TOM) nor components residing in the mitochondrial IMS are involved in this process. Currently, it is unresolved whether the MIM complex has only a receptor-like function or it acts also as an insertase (*Vögtle et al., 2015*). In addition to mediating the membrane integration of multi-span proteins, Mim1 is also involved in the biogenesis of the import receptors Tom20 and Tom70 and therefore the protein is also required for the proper assembly of the TOM complex (*Becker et al., 2008a*; *Dimmer et al., 2012*; *Hulett et al., 2008*; *Lueder and Lithgow, 2009*; *Thornton et al., 2010*; *Waizenegger et al., 2005*). Mim1 is known to interact with Mim2, another protein of the MOM that has a crucial role in the biogenesis of α-helical multispan proteins (*Dimmer et al., 2012*; *Krüger et al., 2017*). Both proteins form a high-molecular-weight complex (MIM complex). They transverse the MOM once and expose their N-terminal domains to the cytosol whereas their C-terminal regions are facing the IMS (*Dimmer et al., 2012*; *Ishikawa et al., 2004*; *Lueder and Lithgow, 2009*; *Waizenegger et al., 2005*).

Considering their multifaceted functions, it is not surprising that the absence of Mim1 and/or Mim2 results in severe growth retardation and multiple cellular defects like hampered assembly of the TOM complex, alteration in mitochondrial morphology, and accumulation of unprocessed mitochondrial precursor proteins (*Dimmer et al., 2012*; *Ishikawa et al., 2004*; *Mnaimneh et al., 2004*; *Popov-Celeketić et al., 2008*; *Waizenegger et al., 2005*). Mim1 and Mim2 are conserved among various fungi but homologues in any other eukaryotes were not identified so far (*Dimmer et al., 2012*; *Ishikawa et al., 2004*; *Otera et al., 2007*; *Waizenegger et al., 2005*). This situation raises the question which factor(s) facilitate the membrane integration of helical MOM proteins in non-fungal organisms.

Recently, a first candidate for such a factor was reported in the parasitic protozoan *Trypanosoma brucei*. It was shown that the integral MOM protein, *p*eripheral *a*rchaic *t*ranslocase of the *o*uter *m*embrane 36 (pATOM36), in analogy to the MIM complex, is involved in the assembly and/or membrane insertion of a small subset of MOM proteins including subunits of the main trypanosomal outer membrane protein translocase (ATOM complex) (*Bruggisser et al., 2017*; *Käser et al., 2016*). However, in contrast to the MIM complex, pATOM36 is also directly required for the inheritance of the single unit mitochondrial genome of trypanosomes, termed kinetoplast DNA (kDNA). A fraction of the protein localises to the tripartite attachment complex (TAC) (*Käser et al., 2016*), which connects the kDNA across the two mitochondrial membranes with the basal body of the flagellum (*Schnarwiler et al., 2014*).

Although pATOM36 and Mim1/2 do not share any sequence or topological similarities (*Figure 1—figure supplement 1*), we wondered whether convergent evolution allowed these unrelated proteins to fulfil similar tasks in the biogenesis of MOM proteins. To address this question, we expressed pATOM36 in yeast cells. Remarkably, introduction of pATOM36 could complement the deletion of *MIM1, MIM2,* or even of both genes. Accordingly, the presence of pATOM36 in the deletion strains could reverse the known alterations resulting from the absence of the MIM complex. Importantly, the reciprocal complementation was also successful and co-expression of Mim1 and Mim2 in *T. brucei* cells ablated for pATOM36 could rescue all phenotypes associated with the MOM protein biogenesis function of pATOM36. Taken together, we present the first reciprocal functional rescue of two evolutionary unrelated mitochondrial biogenesis complexes between eukaryotic supergroups.

## Results

### pATOM36 forms a native-like complex in yeast cells

To better understand the functional relation between yeast Mim1/2 and *T. brucei* pATOM36, we wanted to investigate whether the trypanosomal protein can complement the phenotypes observed in yeast cells lacking the MIM complex. To that aim, plasmids encoding for pATOM36 or its

C-terminally 3xHA-tagged version (pATOM36-HA), as well as an empty plasmid (Ø) as a control, were transformed into wild type (WT), *mim1Δ*, *mim2Δ* or *mim1Δ/mim2Δ* cells. In *T. brucei*, pATOM36 is an integral MOM protein with the C-terminus exposed to the cytosol (*Pusnik et al., 2012*). Blue native (BN)-PAGE analysis has shown that the endogenous protein occurs in two groups of protein complexes of unknown composition with molecular weights of approximately 140–250 kDa and larger than 480 kDa (*Käser et al., 2016*; *Pusnik et al., 2012*).

Initially, we verified that pATOM36-HA can be expressed in the aforementioned yeast strains (*Figure 1—figure supplement 2*). Next, we isolated mitochondria from either control or *mim1Δ/mim2Δ* cells harbouring pATOM36-HA. We observed that the C-terminally HA-tagged pATOM36, similar to the yeast import receptor Tom70, is accessible to added proteinase K in isolated mitochondria, whereas the matrix protein Hep1 was protected as would be expected for intact organelles

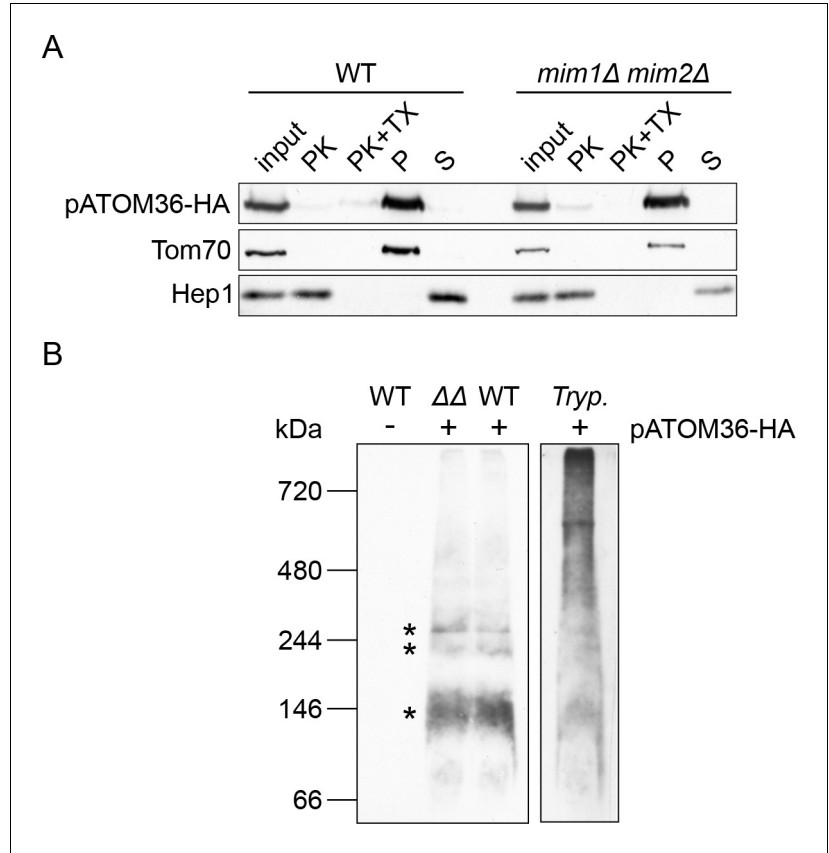

**Figure 1.** pATOM36 forms native-like complexes in the yeast mitochondrial OM. (**A**) Mitochondria isolated from WT or *mim1Δ/mim2Δ* cells expressing pATOM36-HA were left intact or lysed with Triton X-100 (TX) before they were subjected to treatment with proteinase K (PK). Alternatively, other samples were subjected to alkaline extraction followed by separation by centrifugation to pellet (P) and supernatant (S) fractions. All samples were analysed by SDS-PAGE followed by immunodecoration with antibodies against the HA-epitope, the OM receptor protein Tom70, or the matrix soluble protein Hep1. (**B**) Mitochondria were isolated from yeast WT cells transformed with an empty plasmid (-) or from WT and *mim1Δ/mim2Δ* (ΔΔ) cells expressing pATOM36-HA (+). Isolated yeast organelles and mitochondria-enriched fraction from *T. brucei* (Tryp.) cells expressing pATOM36-HA were lysed with 1% digitonin. All samples were then subjected to BN-PAGE followed by immunodecoration with an antibody against the HA-tag. pATOM36-containing complexes are indicated with an asterisk.

DOI: https://doi.org/10.7554/eLife.34488.002

The following figure supplements are available for figure 1:

**Figure supplement 1.** Topologies and protein sequence alignments of Mim1, Mim2 and pATOM36.
DOI: https://doi.org/10.7554/eLife.34488.003
**Figure supplement 2.** pATOM36-HA is expressed in the transformed cells.
DOI: https://doi.org/10.7554/eLife.34488.004

(*Figure 1A*). Alkaline extraction of the isolated organelles showed that pATOM36, as Tom70 but unlike the soluble matrix protein Hep1, was detected in the pellet fraction indicating that it is an integral membrane protein (*Figure 1A*). Finally, a BN-PAGE analysis demonstrated that pATOM36 expressed in yeast forms complexes of similar size to the 140 and 250 kDa complexes observed in *T. brucei* mitochondria (*Figure 1B*). However, the higher molecular weight complex, which likely corresponds to a TAC subcomplex required for kDNA maintenance (*Käser et al., 2016*), was not detected. In summary, these results suggest that pATOM36 expressed in yeast cells behaves essentially identical to the endogenous protein: it is embedded into the MOM with its C-terminus facing the cytosol and it forms oligomeric complexes of ca. 140–250 kDa.

## pATOM36 can replace the MIM complex in yeast

We next asked whether pATOM36 can rescue the growth defect on respiratory carbon sources of *mim1Δ* or *mim2Δ* cells. To that aim, plasmids encoding for pATOM36 or its HA-tagged version, as well as *MIM1* or *MIM2* and an empty plasmid (Ø) as a control, were transformed into wild type, *mim1Δ* and *mim2Δ* strains. The growth of the transformed cells was analysed by drop dilution assays on synthetic fermentative glucose-containing (SD-Leu) and respiratory glycerol-containing media (SG-Leu) at three different temperatures (15°C, 30°C and 37°C). Of note, the expression of pATOM36 and its HA-tagged version did not alter the growth of WT cells. Under all the tested conditions, pATOM36 and pATOM36-HA were able to rescue the growth defect caused by the absence of either Mim1 or Mim2 (*Figure 2A*). Of note, the rescue capacity of pATOM36 was similar to that of Mim1 or Mim2 in the corresponding deletion strains.

These results suggest that pATOM36 is active in yeast cells but it remained unclear whether pATOM36 can function alone or if it requires one of the remaining Mim proteins. To address this question, we monitored the capacity of pATOM36 to rescue the growth retardation of the double deletion *mim1Δ/mim2Δ* cells. We observed that pATOM36 could functionally compensate for the absence of both Mim1 and Mim2, since it was able to rescue the growth defect on non-fermentable carbon sources, a condition which requires fully functional mitochondria (*Figure 2B* and *Figure 2— figure supplement 1*).

The absence of Mim1 and/or Mim2 in yeast cells results in a variety of mitochondrial defects including reduction in the steady-state levels of Mim1/2 substrates like the outer membrane proteins Ugo1, Tom20 and Tom70 (*Dimmer et al., 2012*; *Ishikawa et al., 2004*; *Popov-Celeketić et al., 2008*; *Waizenegger et al., 2005*). We therefore monitored whether expression of pATOM36 restores the reduced levels of these MIM substrates. To that aim we isolated mitochondria from WT and *mim1Δ/mim2Δ* cells transformed with either an empty plasmid or a plasmid encoding pATOM36-HA and monitored the levels of the proteins by immunodecoration. The results indicate that, whereas expression of pATOM36-HA in WT cells did not alter the abundance of the tested proteins or did it only to a minor extent, it did restore the levels of Mim1/2 substrates Tom20 and Tom70 in mitochondria from the double deletion cells (*Figure 3A and B*). Interestingly, the effect of pATOM36 on the levels of Ugo1 was only marginal, suggesting that pATOM36 has preferences to certain MIM substrates.

A further phenotype of cells lacking Mim1/2 is the accumulation of mitochondrial precursor proteins due to hampered assembly of the TOM complex (*Ishikawa et al., 2004*; *Mnaimneh et al., 2004*; *Waizenegger et al., 2005*). To test whether pATOM36 is able to reverse this situation, we obtained whole cell lysates from the cells described above. As can be seen in *Figure 3C*, the presence of pATOM36-HA in the deletion strains completely eliminated the appearance of the precursor form of mitochondrial Hsp60. The presence of pATOM36-HA in the deletion cell lines resulted also in enhanced levels of Tom40 whereas the amounts of aconitase (Aco1) were not affected (*Figure 3C*).

These results suggest that the function of the MIM complex in TOM complex assembly can be replaced by pATOM36. To substantiate this assumption, we used digitonin-solubilised mitochondria, which were isolated from control and deletion strains, and analysed them by BN-PAGE. To detect the TOM complex, the corresponding immunoblots were probed with antibodies against either Tom40 or Tom22. Of note, pATOM36-HA did not affect the assembly of the TOM complex in WT cells (*Figure 3D*). As expected, in the absence of Mim1/2, a dramatic reduction in the amount of assembled TOM complex and an appearance of an unassembled Tom40-containing species can be

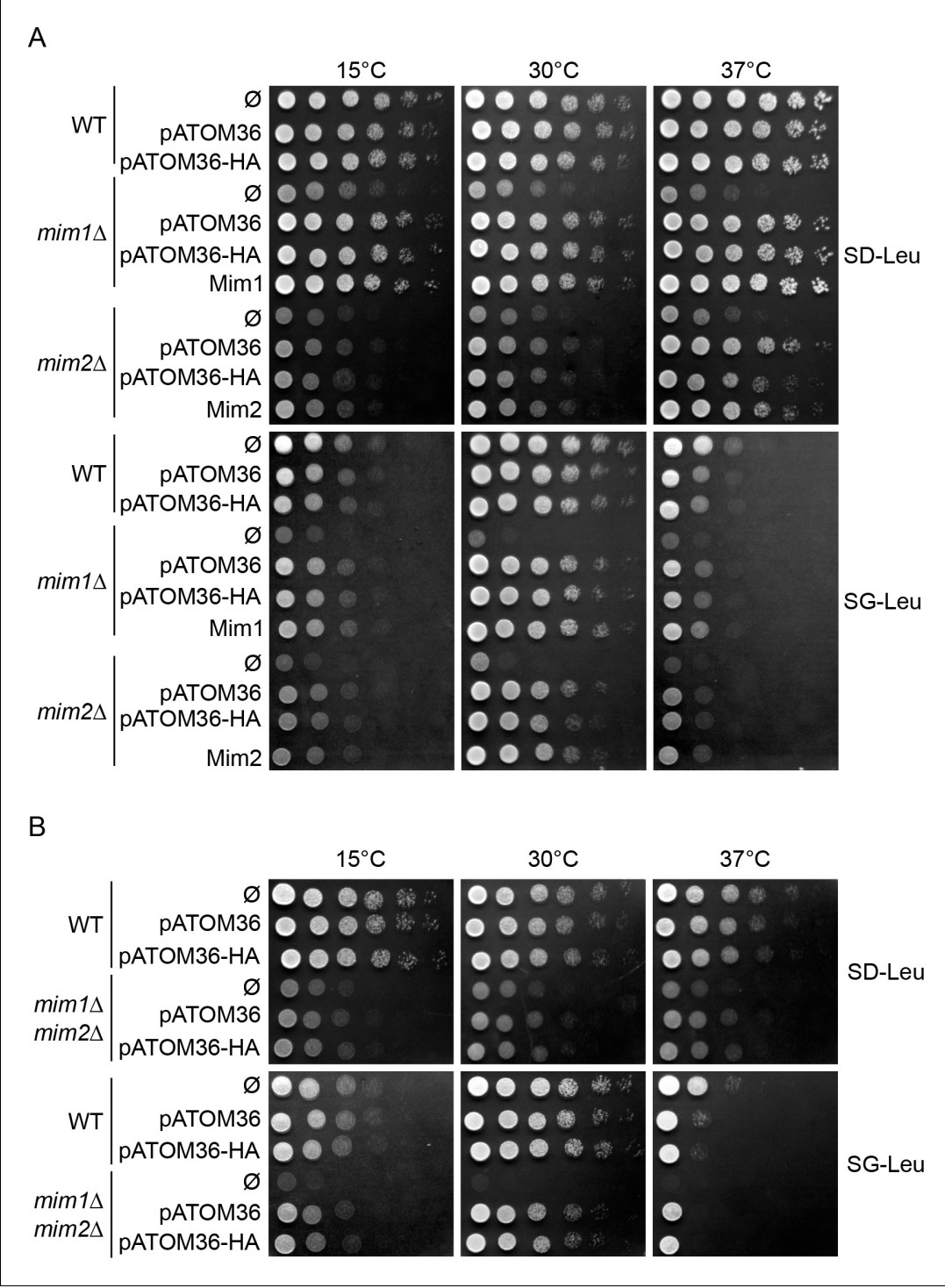

**Figure 2.** pATOM36 rescues the growth defects of cells lacking Mim1, Mim2 or both. (**A**) The indicated strains transformed with an empty plasmid (Ø) or with a plasmid expressing pATOM36 or its HA-tagged variant were tested at three different temperatures by drop-dilution assay for growth on synthetic medium containing either glucose (SD-Leu) or glycerol (SG-Leu). For comparison, plasmid-encoded Mim1 or Mim2 were transformed into *mim1Δ* or *mim2Δ* cells, respectively. All dilutions are in fivefold increment. (**B**) Cells deleted for both *MIM1* and *MIM2* (*mim1Δ/mim2Δ*) were transformed with the empty plasmid (Ø) or a plasmid encoding either native pATOM36 or pATOM36-HA. Transformed cells were analysed by drop-dilution assay at the indicated temperatures on synthetic medium containing either glucose (SD-Leu) or glycerol (SG-Leu). All dilutions are in fivefold increment.

*Figure 2 continued on next page*

*Figure 2 continued*

DOI: https://doi.org/10.7554/eLife.34488.005

The following figure supplement is available for figure 2:

**Figure supplement 1.** pATOM36 rescues the growth defect of mim1Δmim2Δ cells.

DOI: https://doi.org/10.7554/eLife.34488.006

observed. Strikingly, these alterations completely disappeared upon the introduction of pATOM36-HA into these cells (*Figure 3D*).

To investigate the specificity of the complementation by pATOM36, we asked whether it can functionally replace another import factor that mediates the biogenesis of other MOM proteins. To that goal, pATOM36 was introduced into cells lacking Mas37/Sam37, a subunit of the TOB/SAM complex that facilitates membrane integration of β-barrel proteins and the TOM subunit Tom22 (*Chan and Lithgow, 2008*; *Dukanovic et al., 2009*; *Wiedemann et al., 2003*). *Figure 3—figure supplement 1* shows that pATOM36 could revert neither the drop in the steady-state levels of the TOB complex and its altered assembly behaviour nor the reduced levels of either the β-barrel proteins Tom40 and Porin or the single-span protein Tom22. Thus, the effect of pATOM36 is specific for MIM substrates. These findings further support the notion that the single-span protein Tom22 follows an import pathway that is distinct from that taken by the signal-anchored subunits Tom20 and Tom70.

Previous reports suggested that Tom70 works together with Mim1 in the biogenesis of multi-span helical MOM proteins (*Becker et al., 2011*; *Papic et al., 2011*). To test whether pATOM36 can also interact with Tom70, we utilised a recombinant protein composed of the cytosolic domain of Tom70 fused to GST moiety (GST-Tom70). When this protein was incubated with newly synthesised radiolabelled pATOM36, or with Mim1 as a control, we observed a specific binding to both proteins (*Figure 3E*). Although we cannot exclude the possibility that Tom70, as an import receptor for MOM proteins, recognises Mim1 and pATOM36 as substrates, it can be envisaged that, similarly to Mim1, pATOM36 can also cooperate with Tom70 in the biogenesis of MOM proteins.

The aforementioned results indicate that pATOM36 can compensate for the loss of the MIM machinery. To demonstrate directly a role of pATOM36 in protein import into the outer membrane of yeast mitochondria, we performed in vitro import assays. To that aim, we tested whether the presence of pATOM36 in mitochondria lacking the MIM complex can rescue the reduced import capacity of the MIM substrates Tom20 and Ugo1 observed for these organelles. To monitor the import efficiency of radiolabelled Tom20 into isolated organelles, we employed an established assay based on the formation of a proteolytic fragment of an N-terminally extended variant of Tom20 (*Ahting et al., 2005*). This assay clearly demonstrated that the presence of pATOM36 is sufficient to improve dramatically the capacity of organelles lacking Mim1/2 to import radiolabelled Tom20 molecules (*Figure 4A*). Along the same line, the assembly of newly synthesised Tom20 molecules into pre-existing TOM complexes was markedly improved when pATOM36 was present in mitochondria lacking the MIM complex (*Figure 4B*). Similarly to its minor effect on the steady state levels of Ugo1, the presence of pATOM36 did not improve the capacity of isolated mitochondria to import radiolabelled Ugo1 (*Figure 4C*). As a control, we checked the effect of pATOM36 on the import of proteins that are not known as MIM substrates like the matrix-targeted model protein pSu9-DHFR or the MOM tail-anchored protein Fis1. In both cases, we did not observe altered import upon expression of pATOM36 (*Figure 4D and E*). Collectively, pATOM36 can support the biogenesis of MIM substrates but appears to have preferences to certain ones.

Finally, we tested whether the trypanosomal protein is able to rescue the mitochondrial fragmentation that is observed in cells lacking Mim proteins. To that goal, we transformed a plasmid encoding pATOM36 into WT, *mim1Δ*, *mim2Δ*, or *mim1Δ/mim2Δ* cells expressing mitochondrial targeted GFP (mito-GFP). Analysis of mitochondria from the resulting cell lines by fluorescence microscopy revealed that pATOM36 is able to revert the mitochondrial fragmentation observed in cells lacking Mim1 and/or Mim2 to the tubular-like morphology of organelles in control cells (*Figure 5A and B*).

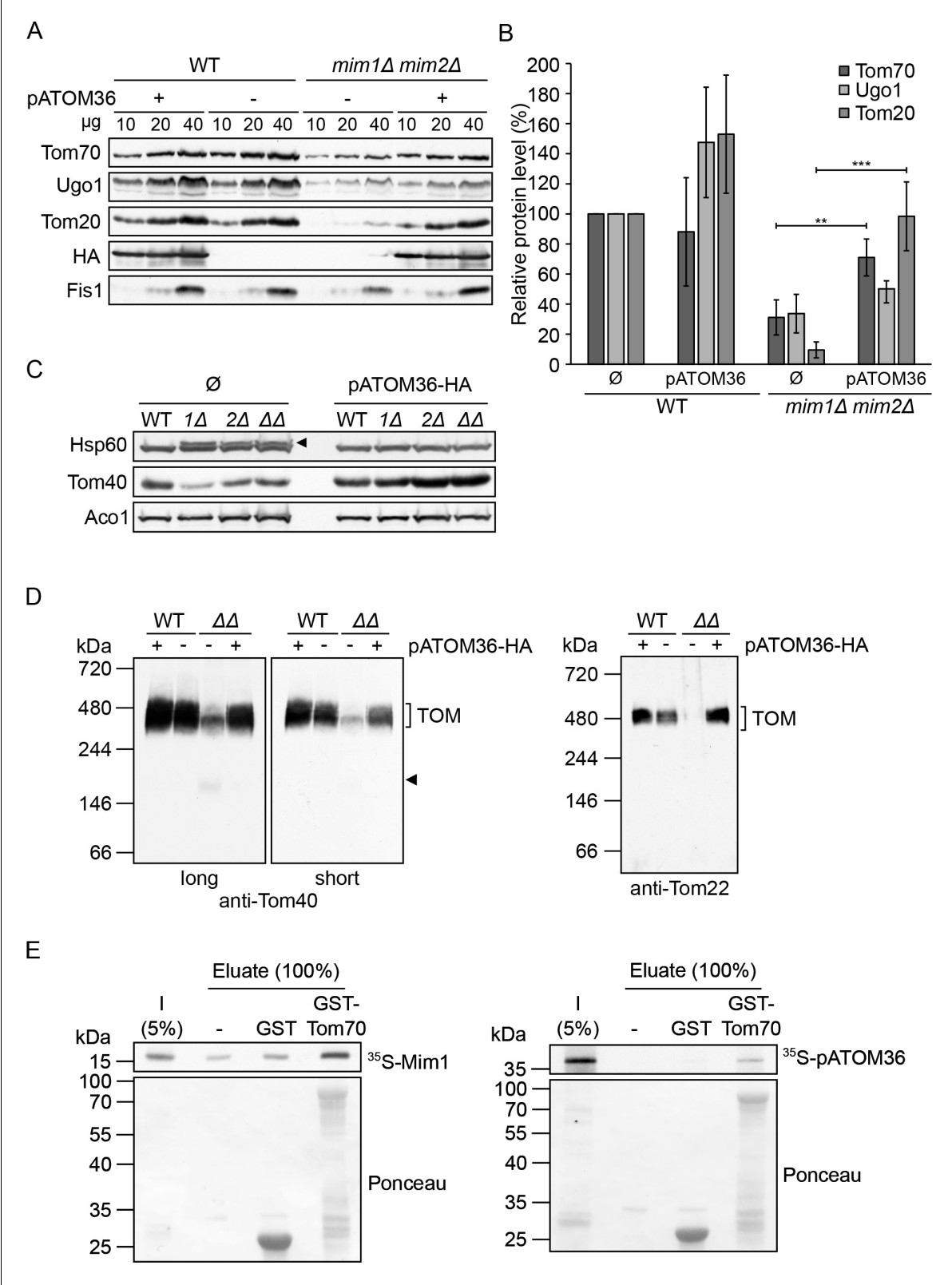

**Figure 3.** pATOM36 can compensate for the reduced steady state levels and assembly defects in cells lacking both Mim1 and Mim2. (**A**) Mitochondria were isolated from WT or *mim1Δ/mim2Δ* cells transformed with either an empty plasmid (-) or with a plasmid encoding pATOM36-HA (+). The specified amounts were analysed by SDS-PAGE and immunodecoration with antibodies against either the indicated mitochondrial proteins or the HA-tag. (**B**) The intensity of the bands from three independent experiments such as those presented in (**A**) was monitored. The amounts of Tom70, Ugo1 and

*Figure 3 continued on next page*

*Figure 3 continued*

Tom20 in the various mitochondria samples are presented as mean percentage of their levels in control organelles (WT+ Ø). The levels of Fis1 were taken as loading control. Error bars represent ± SD. **p≤0.005, ***p≤0.0005. (C) Whole cell lysates were obtained from WT, mim1Δ (1Δ), mim2Δ (2Δ), or the double deletion mim1Δ/mim2Δ (ΔΔ) cells transformed with either an empty plasmid (Ø) or with a plasmid encoding pATOM36-HA. Samples were analysed by SDS-PAGE and immunodecoration with antibodies against the indicated mitochondrial proteins. The precursor form of mitochondrial Hsp60 is indicated with an arrowhead. (D) The mitochondria described in (A) were solubilised in a buffer containing 1% digitonin and then analysed by BN-PAGE followed by western blotting. The membranes were immunodecorated with antibodies against the TOM subunits, Tom40 (long and short exposures) and Tom22. The TOM complex is signposted. A Tom40-containing low molecular weight complex is indicated with an arrowhead. (E) Mim1 and pATOM36 interact directly with Tom70. Radiolabelled Mim1 or pATOM36 (input, I) were incubated with glutathione beads (-) or with beads that were pre-bound to recombinant GST alone or to GST fused to the cytosolic domain of Tom70 (GST-Tom70). After washing, bound material was eluated and proteins were analysed by SDS–PAGE followed by blotting onto a membrane, and detection with either autoradiography (upper panel) or Ponceau staining (lower panel).

DOI: https://doi.org/10.7554/eLife.34488.007

The following source data and figure supplement are available for figure 3:

**Source data 1.** pATOM36 can compensate for the reduced steady state levels in cells lacking both Mim1 and Mim2.

DOI: https://doi.org/10.7554/eLife.34488.009

**Figure supplement 1.** pATOM36-HA does not rescue biogenesis defects in mas37Δ cells.

DOI: https://doi.org/10.7554/eLife.34488.008

## Mim1/2 form a native-like MIM complex in trypanosomes

Observing the rescue capacity of pATOM36 in yeast cells, we asked whether the functional similarity between Mim1/2 and pATOM36 allows the yeast proteins to replace the function pATOM36 has in the biogenesis of trypanosomal MOM proteins. To that end, we constructed a plasmid for the co-expression of myc-tagged Mim1 and HA-tagged Mim2 in *T. brucei* (*Figure 6A*). Next, this plasmid was introduced into a cell line allowing controlled ablation of pATOM36. In these cells, addition of tetracycline simultaneously initiates the RNAi-mediated degradation of the pATOM36 mRNA as well as the expression of the tagged Mim1 and Mim2.

Subcellular fractionation of induced cells showed that both proteins are expressed and, like the mitochondrial marker protein ATOM40, they are exclusively localised in the mitochondrial fraction (*Figure 6B*, top panels). Alkaline extraction of the latter revealed that, as the endogenous proteins in yeast, both Mim1 and Mim2 are recovered in the pellet, together with the integral membrane protein ATOM40, whereas the soluble protein CytC was present in the supernatant (*Figure 6B*, lower panels). To monitor whether Mim1 and Mim2 are inserted into the membrane in their native orientation, mitochondria-enriched fractions were treated with proteinase K. This treatment resulted for both proteins in the formation of protease-resistant C-terminal fragments (*Figure 6C*). Thus, Mim1 and Mim2 acquired their native topology in *T. brucei* mitochondria with their N-terminus exposed to the cytosol and the C-terminus located in the IMS. Mim1 and Mim2 of yeast cells form a complex of approx. 200 kDa (*Dimmer et al., 2012*; *Ishikawa et al., 2004*; *Waizenegger et al., 2005*). BN-PAGE shows that similar complexes of ca. 230 kDa, which contain both Mim1-myc and Mim2-HA, could be detected in *T. brucei* (*Figure 6D*). Importantly, these complexes migrated similarly to complexes harbouring Mim1-HA and Mim2-HA of yeast mitochondria (*Figure 6E*). The slightly higher molecular weight than that observed for native complexes in yeast can be explained by the fact that both proteins are tagged. Thus, expression of Mim1 and Mim2 results in a native-like MIM complex in mitochondria from *T. brucei*.

## The MIM complex can replace the protein biogenesis function of pATOM36 in *T. brucei*

The next question we addressed was whether the MIM complex can take over the function of pATOM36. Ablation of pATOM36 has been shown to cause a growth arrest. Due to its dual function the lack of pATOM36 does not only interfere with the assembly and/or insertion of MOM proteins but it also prevents assembly of the TAC, which causes loss of the kDNA (*Figure 7A*) (*Käser et al., 2016*). Interestingly, introducing Mim1/2 into the pATOM36-depleted cells could not prevent the loss of kDNA but it did cause a milder growth phenotype (*Figure 7A*). When mitochondrial proteins from pATOM36-depleted cells expressing Mim1/2 were analysed, we observed that the steady-state levels of the ATOM complex subunits ATOM46, ATOM19, and ATOM14, all of which are greatly

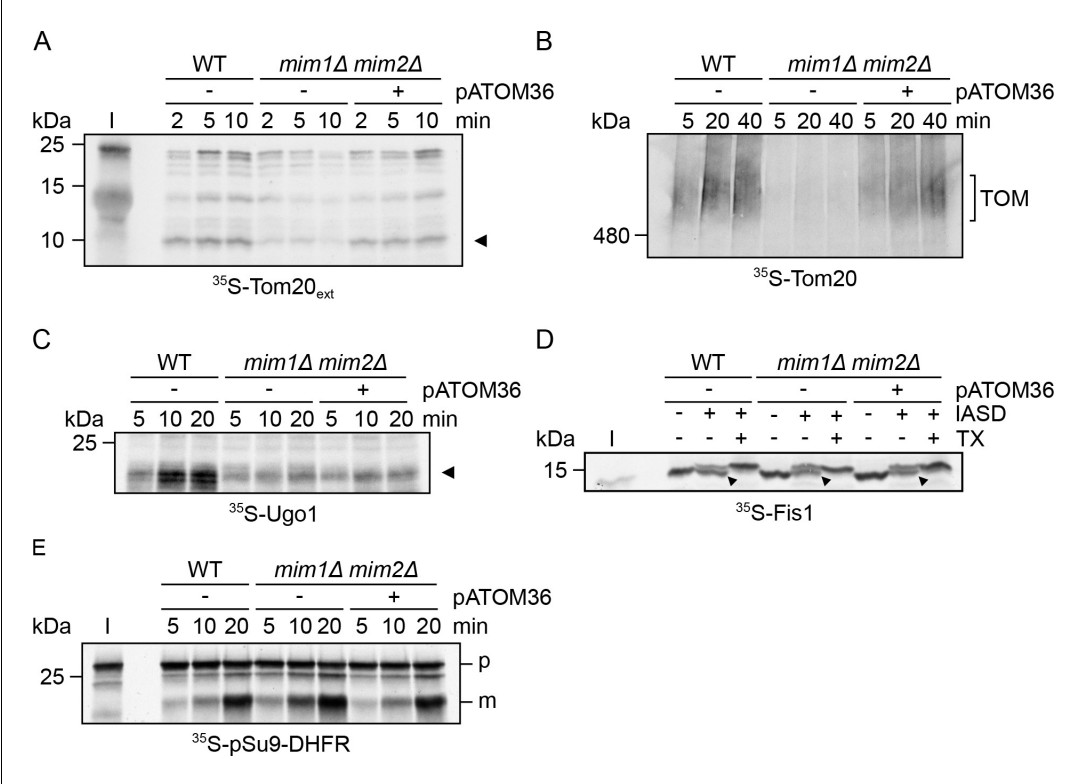

**Figure 4.** pATOM36 can rescue some of the import defects of cells lacking the MIM complex. (**A**) Mitochondria were isolated from WT cells transformed with an empty plasmid (WT-) or from *mim1Δ/mim2Δ* cells transformed with either an empty plasmid (-) or with a plasmid encoding pATOM36-HA (+). Radiolabelled Tom20ext molecules (5% input, I) were incubated with the indicated isolated organelles for the specified time periods. Then, mitochondria were treated with PK and analysed by SDS-PAGE and autoradiography. A proteolytic fragment of Tom20ext, which reflects correct membrane integration, is indicated by an arrowhead. (**B**) Radiolabelled Tom20 was incubated with isolated mitochondria as in (**A**). At the end of the import reactions, mitochondria were solubilised with 0.2% digitonin and samples were analysed by BN-PAGE followed by autoradiography. The migration of Tom20 molecules assembled into the TOM complex is indicated. (**C**) Radiolabelled Ugo1 was incubated with isolated mitochondria as in (**A**). Then, mitochondria were treated with trypsin and analysed by SDS-PAGE and autoradiography. A proteolytic fragment of Ugo1, which reflects correct membrane integration, is indicated by an arrowhead. (**D**) Radiolabelled Fis1-TMC (5% input, I) was incubated with isolated mitochondria as in (**A**). Then, mitochondria were subjected to an IASD assay, re-isolated and analysed by SDS-PAGE and autoradiography. Bands representing correctly integrated Fis1-TMC are marked by an arrowhead. (**E**) Radiolabelled pSu9-DHFR (5% input, I) was incubated with isolated mitochondria as in (**A**). Then, mitochondria were re-isolated and analysed by SDS-PAGE and autoradiography. The precursor and mature forms are indicated by p and m, respectively.

DOI: https://doi.org/10.7554/eLife.34488.010

reduced in the absence of pATOM36, were restored (*Figure 7B*) (*Käser et al., 2016*). Furthermore, not only the abundance of the ATOM subunits was back to normal levels, but also the subunits were incorporated into the high-molecular-weight ATOM complexes. Of note, in the cell lines complemented by the MIM complex the ATOM subunit complexes were shifted to a slightly higher molecular weight (*Figure 8A*). Moreover, complementation of the ATOM40-containing complexes was somewhat incomplete, since the 200 kDa ATOM40 complexes that accumulate after ablation of pATOM36 were still visible (*Figure 8A*).

It has previously been described that ablation of pATOM36 in trypanosomes, reminiscent to deletion of the MIM complex in yeast, causes a condensation of the network-like structure of the trypanosomal mitochondrion (*Bruggisser et al., 2017*) (*Figure 8B*, left panel). The immunofluorescence analysis in the right panel of *Figure 8B* indicates that in the presence of the MIM complex also this phenotype is reversed and the wild type morphology of the mitochondrion is fully restored. Hence, similarly to the rescue capacity of pATOM36 in yeast cells, Mim1/2 can replace the function of endogenous pATOM36 in MOM protein biogenesis in trypanosomes.

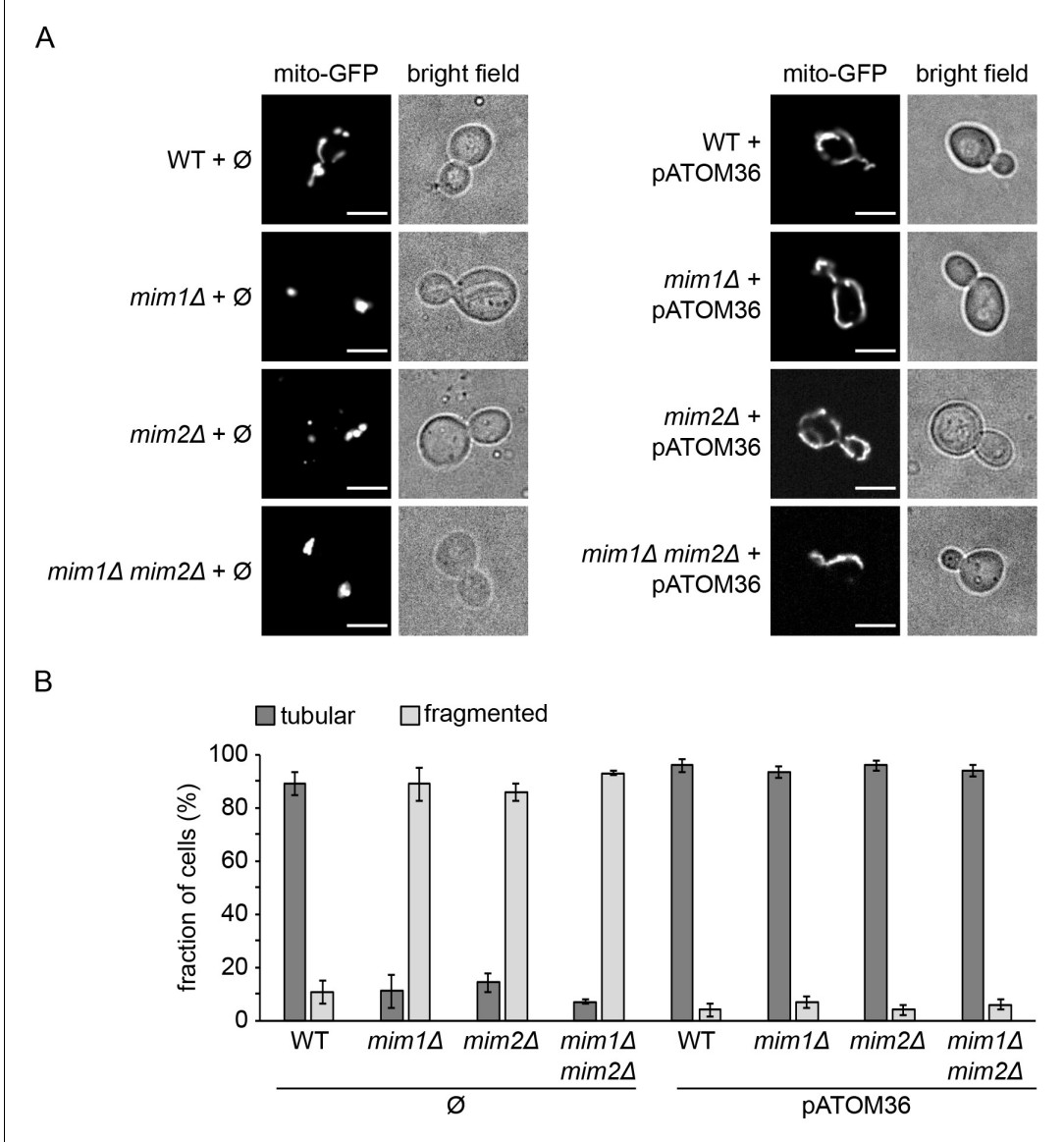

**Figure 5.** *mim1Δ* and *mim2Δ* cells expressing pATOM36 do not show altered mitochondrial morphology. (**A**) WT, *mim1Δ*, *mim2Δ*, and *mim1Δ/mim2Δ* cells harbouring mitochondria-targeted GFP (mito-GFP) were transformed with either an empty plasmid (Ø) as a control (left panels) or a plasmid encoding pATOM36 (right panels). Cells were analysed by fluorescence microscopy and representative images of the predominant morphology for each strain are shown. Scale bar, 5 µm. (**B**) Statistical analysis of the cells described in (**A**). Average values with standard deviation bars of three independent experiments with at least n = 100 cells in each experiment are shown.

DOI: https://doi.org/10.7554/eLife.34488.011

The following source data is available for figure 5:

**Source data 1.** *mim1Δ* and *mim2Δ* cells expressing pATOM36 have normal mitochondrial morphology

DOI: https://doi.org/10.7554/eLife.34488.012

## Complementing the biogenesis function of pATOM36 requires both Mim1 and Mim2

When we transfected the *T. brucei* pATOM36-RNAi cell line with distinct plasmids encoding myc-tagged Mim1 and HA-tagged Mim2 we obtained also clones that mainly expressed either Mim1-myc or Mim2-HA while the other Mim subunit was expressed only in residual amounts (*Figure 8— figure supplement 1*). In the cell line that mainly expresses Mim1-myc, the protein is found in a complex of approximately 440 kDa (*Figure 8—figure supplement 1A*, middle panel), whereas in the cell

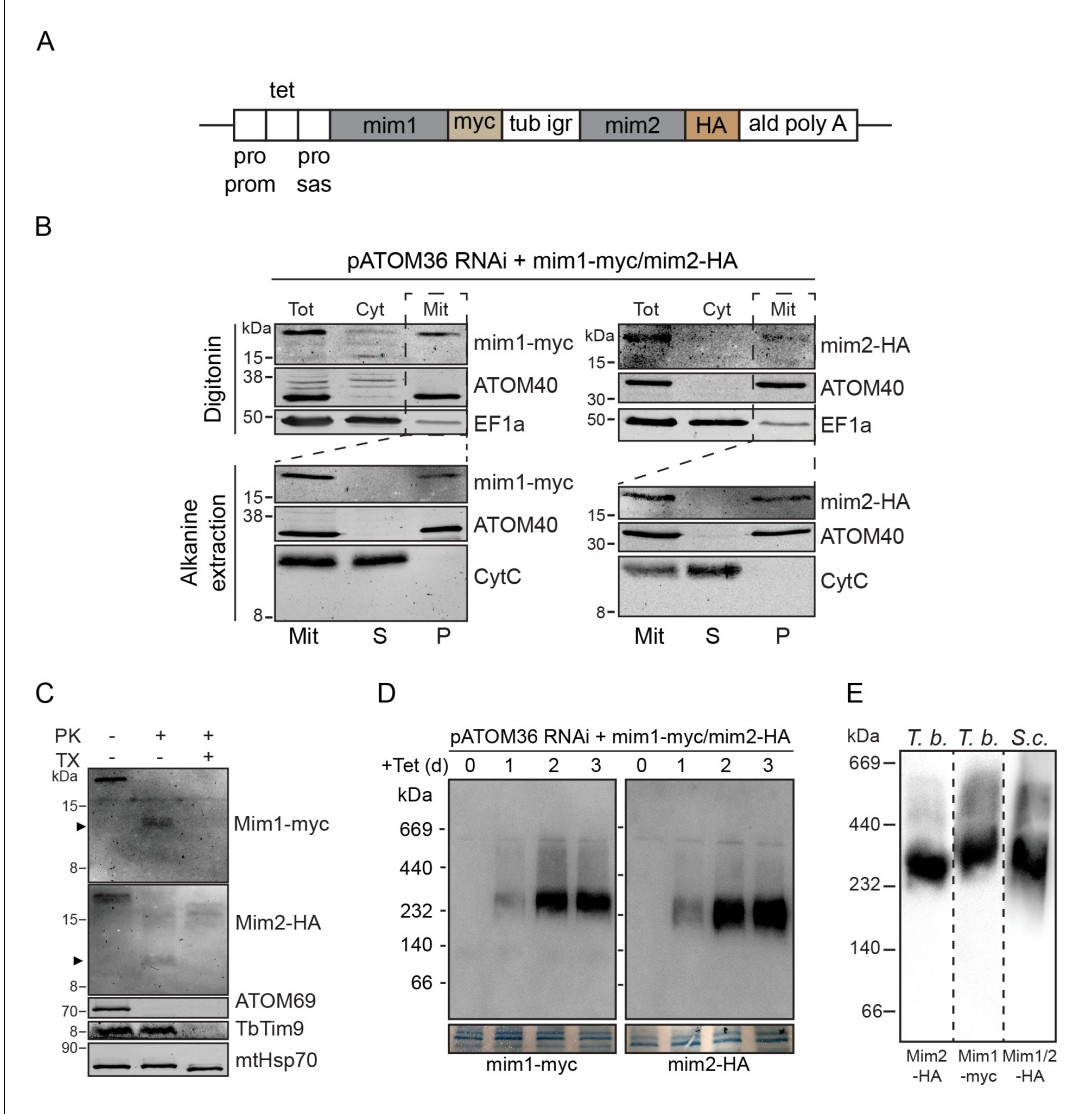

**Figure 6.** Yeast Mim1 and Mim2 form a high-molecular-weight complex in mitochondria of *T. brucei*. (**A**) Schematic representation of the insert of the pLew100-based vector that allows tetracycline-inducible expression of C-terminally myc-tagged Mim1 and HA-tagged Mim2 in *T. brucei*. Pro prom, procyclin promotor; tet, tetracycline operator; pro sas, procycline splice acceptor site; tub igr, α- and β-tubulin intergenic region; ald polyA, 3'-UTR of the aldolase gene. (**B**) Top panels: immunoblot analysis of whole cells (Tot), soluble (Cyt) and digitonin-extracted mitochondria-enriched pellet (Mit) fractions of a tetracycline-inducible pATOM36-RNAi cell line expressing Mim1-myc and Mim2-HA. Duplicate blots were analysed for the expression of Mim1-myc (left panels) and Mim2-HA (right panels). ATOM40 and EF1a serve as mitochondrial and cytosolic markers, respectively. Bottom panels: Alkaline extraction of the mitochondria-enriched fraction (Mit) shown in the top panels. The pellet (**P**) and the supernatant (**S**) fractions corresponding to integral membrane and soluble proteins, respectively, were analysed by SDS-PAGE and immunodecoration. ATOM40 and CytC serve as markers for integral and peripheral membrane proteins, respectively. (**C**) Mitochondria-enriched fractions of the same cell line describe in (**B**) were left intact or lysed with Triton X-100 (TX) before they were subjected to treatment with proteinase K (PK). All samples were analysed by SDS-PAGE followed by immunodecoration with antibodies against myc and HA tags, the OM protein ATOM69, the IMS protein TbTim9, or the matrix protein mtHsp70. Note that mtHsp70 contains a folded core, which is protease resistant. A proteolytic fragment of Mim1 and Mim2 is indicated with an arrowhead. (**D**) Duplicate immunoblots from BN-PAGE analysis of mitochondria-enriched fractions of the same cell line describe in (**B**) were probed for Mim1-myc (left panels) and Mim2-HA (right panels). Sections of the coomassie-stained gels serve as loading control. (**E**) Immunoblots of a BN-PAGE analysis of mitochondria-enriched fractions of the *T. brucei* (*T.b.*) cell line simultaneously expressing myc-tagged Mim1 (Mim1-myc) and HA-tagged Mim2 (Mim2-HA) and isolated yeast (*S.c.*) mitochondria simultaneously expressing HA-tagged versions of Mim1 and Mim2. The immunoblots are probed with antibodies against HA- or myc-tag.

DOI: https://doi.org/10.7554/eLife.34488.013

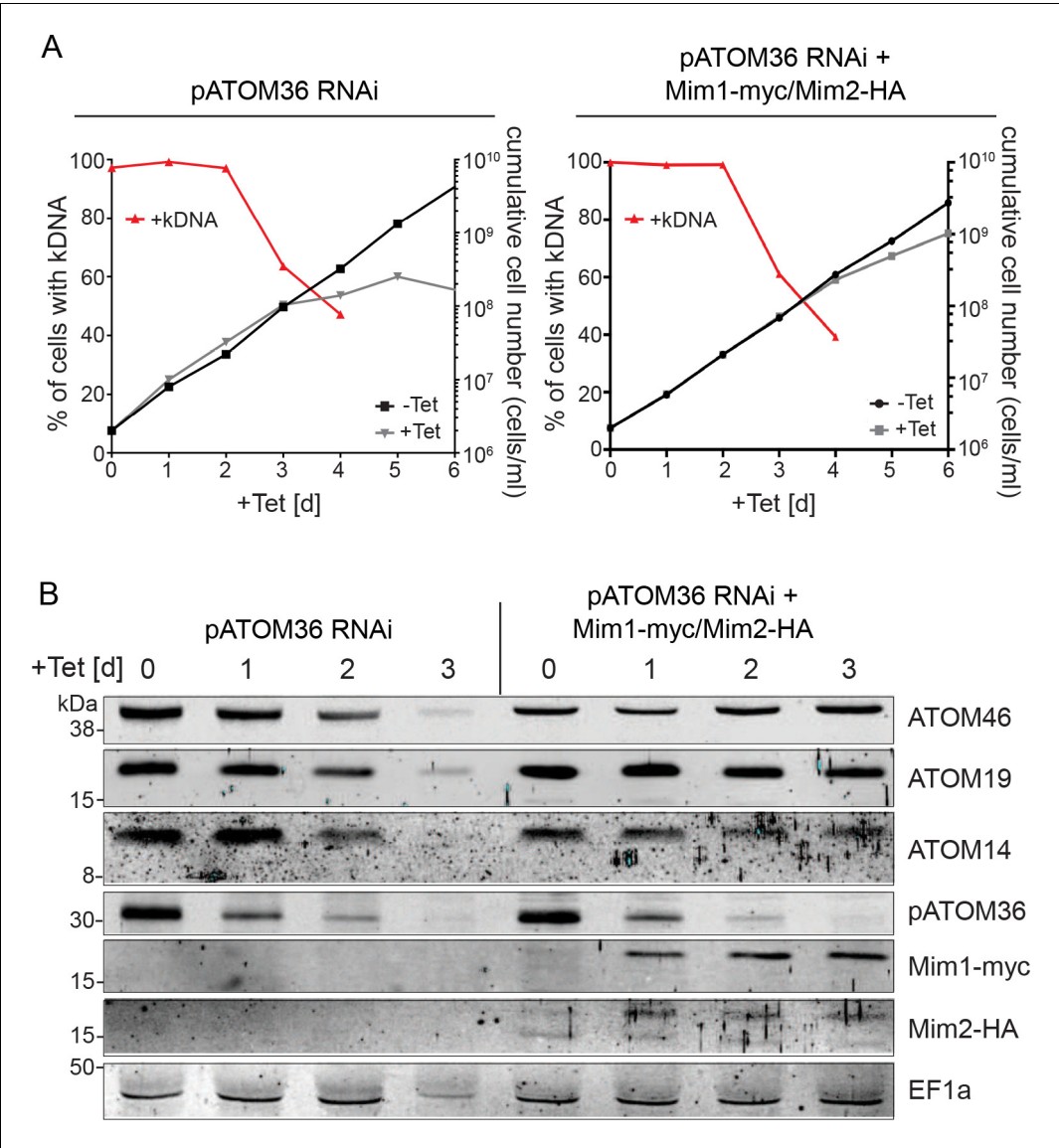

**Figure 7.** Yeast Mim1 and Mim2 complement the mitochondrial OM biogenesis phenotype of *T. brucei* cells ablated for pATOM36. (**A**) Left panel: growth in the presence and absence of tetracycline (black and grey lines, respectively) and loss of kDNA (red line) in the presence of tetracycline of the pATOM36-RNAi parent cell line. Right panel: as in the left but the analysis was done for the pATOM36-RNAi cell line that co-expresses Mim1-myc and Mim2-HA. (**B**) Whole cell lysates from the cell lines as in (**A**) were obtained after the indicated time of induction. Proteins of these samples were analysed by SDS-PAGE and immunodecoration with the indicated antibodies. ATOM46, ATOM19 and ATOM14 are subunits of the ATOM complex. Cytosolic EF1a serves as a loading control.

DOI: https://doi.org/10.7554/eLife.34488.014

line preferentially expressing Mim2-HA this protein is present in a complex of approximately 230 kDa (*Figure 8—figure supplement 1B*, bottom panel). These complexes are of either higher (Mim1-myc) or similar molecular weights (Mim2-HA) to the one that is formed when both proteins are expressed in similar amounts (*Figure 6D and E*). Most importantly, both cell lines show a strong deficiency of ATOM complex assembly (*Figure 8—figure supplement 1*, top panels) and a growth arrest (*Figure 8—figure supplement 1*, bottom graphs) that are indistinguishable from the parent pATOM36-RNAi cell line (*Figure 8A*, left panels and *Figure 7*, left graph, respectively). This indicates that expression of Mim1 or Mim2 alone cannot complement for the protein biogenesis

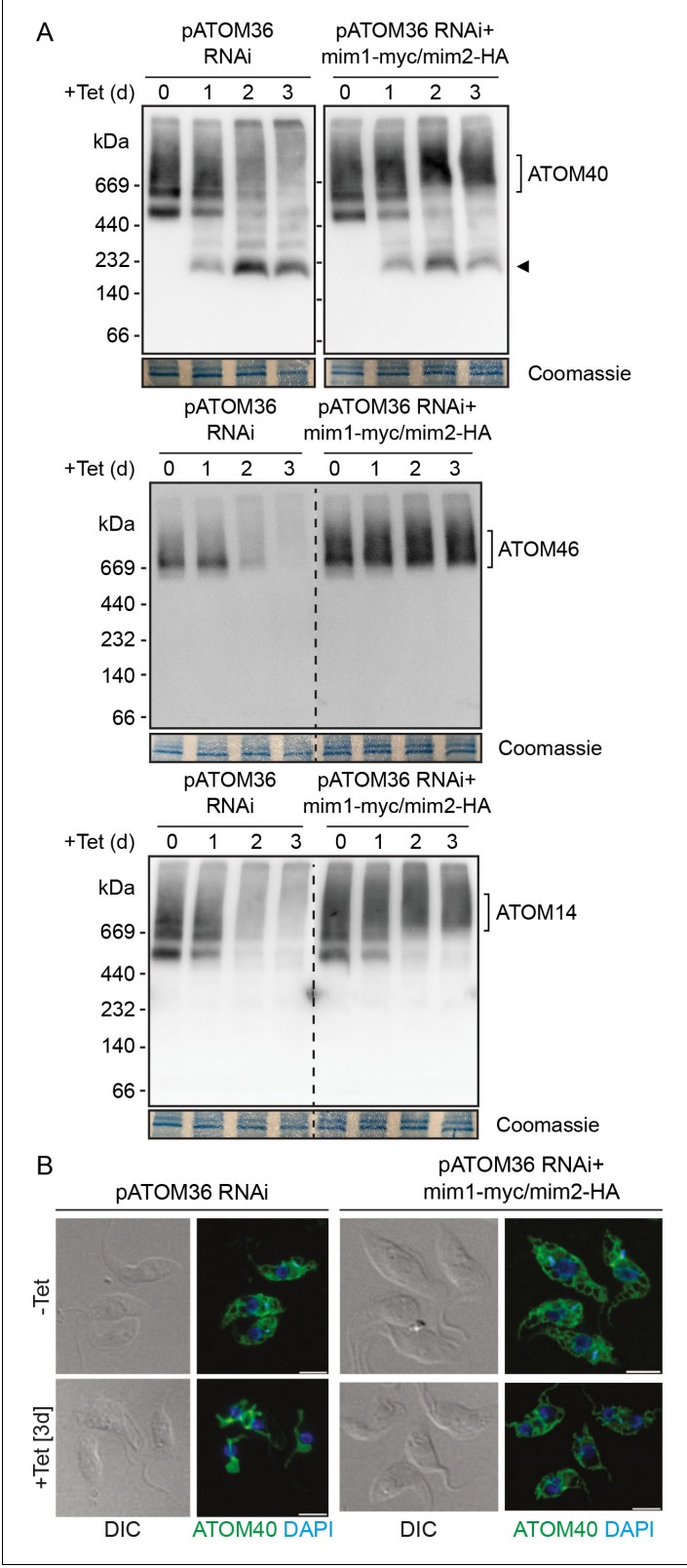

**Figure 8.** Mim1 and Mim2 rescue the assembly defect of the ATOM complex and the altered mitochondrial morphology in cells lacking pATOM36. (**A**) Mitochondria-enriched fractions from the cell lines as in *Figure 7A* were obtained after the indicated time of induction. Samples were analysed by BN-PAGE followed by immunodecoration with antibodies against the indicated subunits of the ATOM complex. The migration of the

*Figure 8 continued on next page*

*Figure 8 continued*
ATOM complex is signposted. Sections of the coomassie-stained gels serve as loading controls. Arrowhead indicates an ATOM40-containing lower molecular weight complex. (B) Left images: Immunofluorescence analyses of mitochondrial morphology in the pATOM36 RNAi cell line after 0 or 3 days of induction. Right images: as in the left panels but the analysis was performed with the RNAi cell line co-expressing Mim1-myc and Mim2-HA. ATOM40 is shown in green and DAPI-stained DNA is shown in blue. DIC, differential interference contrast. Scale bar, 5 μm.
DOI: https://doi.org/10.7554/eLife.34488.015
The following figure supplement is available for figure 8:

**Figure supplement 1.** Complementing the biogenesis function of pATOM36 requires both Mim1 and Mim2.
DOI: https://doi.org/10.7554/eLife.34488.016

phenotype caused by the lack of pATOM36. Furthermore, these results suggest that successful complementation requires similar amounts of Mim1 and Mim2.

## Discussion

Our study shows that the MIM complex of yeast, consisting of Mim1 and Mim2, and trypanosomal pATOM36 have identical functions, even though they do not share sequence similarity, the same membrane topology, or a similar size (*Figure 1—figure supplement 1*). This conclusion is based on the stringent criteria that the two proteins can replace each other in reciprocal complementation experiments. The only major limitation is that the role of pATOM36 in mitochondrial DNA inheritance in trypanosomes cannot be carried out by the MIM complex, which is expected since unlike the MIM complex pATOM36 has a dual function (*Käser et al., 2016*).

The reciprocal complementation is surprising because yeast belongs to the eukaryotic supergroup of the Opisthokonts whereas trypanosomes are Excavates (*Burki, 2014*). Thus, except for being eukaryotes the two systems are essentially unrelated. The most parsimonious explanation for the observed phylogenetic distribution of the two functional analogues is that the MIM complex and pATOM36 evolved after the eukaryotic supergroups were already established. Moreover, the observation that within the Opisthokonts the MIM complex is restricted to fungi suggests that it evolved only after the divergence of the ancestors of fungi and metazoans. Thus, the last eukaryotic common ancestor (LECA) likely did not contain the MIM complex, pATOM36 or any other functional analogue of these proteins. This assumption is in line with the notion that LECA had a much simpler MOM protein import system consisting possibly only of a Tom40-like β-barrel protein (*Dolezal et al., 2006*; *Mani et al., 2016*), whose integration into the MOM is mediated by the TOB/SAM complex, the core subunit of which, Tob55/Sam50, is conserved in all eukaryotes (*Dolezal et al., 2006*; *Gentle et al., 2004*; *Kozjak et al., 2003*; *Paschen et al., 2003*). During evolution, additional subunits that are anchored in the membrane by α-helices joined the TOM complex to increase its specificity and efficiency. This scenario is supported by the fact that the TOM complexes of yeast, plants and trypanosomes, representatives of three different eukaryotic supergroups, contain three distinct evolutionary unrelated pairs of protein import receptors (*Mani et al., 2015*; *Mani et al., 2016*). The appearance of the new TOM subunits required the evolution of a system, such as the MIM complex or pATOM36, that facilitates their assembly with Tom40.

Interestingly, the capacity of pATOM36 expressed in yeast cells to support the biogenesis of MIM substrates is variable with the import receptor Tom20 as the most favourable substrate and the fusion-modulator Ugo1 as the least favourable one. Ugo1 is a carrier-like protein with several TMSs that lack clear homologues in higher eukaryotes. Hence, one can speculate that pATOM36 cannot deal with it well since there are no similar substrates in the MOM of *T. brucei*.

Both Mim1 and Mim2 as well as pATOM36 occur in protein complexes of unknown composition. The successful complementation experiments together with the fact that they form complexes of similar sizes when expressed in the heterologous systems strongly suggest that these complexes do not contain any additional proteins. Their ability for reciprocal rescue also suggests that their essential function does not require any further proteins since it is very unlikely that such factors would be present in the other species.

It is established that pATOM36 and Mim1/Mim2 are integral MOM proteins. However, whereas Mim1 and Mim2 have each a single TMS with the N-terminus facing the cytosol, the topology of pATOM36 is largely unknown (*Figure 1—figure supplement 1*). It has been demonstrated by antibody shift experiments that the C-terminus of pATOM36 is exposed to the cytosol (*Pusnik et al., 2012*), but depending on the prediction programs the protein is postulated to have either one, two or even three TMSs (*Käser et al., 2016*). While Mim1/Mim2 and pATOM36 do not share sequence similarity and also have different molecular weights (Mim1, 13 kDa; Mim2, 11 kDa; pATOM36, 36 kDa), they all have GxxxG(A) motifs within their putative TMSs (*Figure 1—figure supplement 1*), which is in line with their oligomeric quaternary structures. It has recently been shown by electrophysiological experiments that Mim1, on its own or in complex with Mim2, can form a cation-selective channel (*Krüger et al., 2017*). Should this channel activity of Mim1 be functionally relevant, we would expect pATOM36 to form also a pore.

Presently, it is unclear whether the convergent evolution of the MIM complex and pATOM36 demonstrated in the present study, resulted in a similar 3D-structure of the two oligomers. Should this be the case, the two complexes may independently have evolved the same mechanisms to perform the equivalent functions. Alternatively, it cannot be excluded that they use structurally different solutions resulting in different mechanisms that nevertheless allow them to carry out the same functions.

There is evidence that both the yeast MIM complex as well as trypanosomal pATOM36 mediate assembly of already integrated MOM proteins and at least for some substrates also the insertion process itself (*Becker et al., 2008a*; *Becker et al., 2011*; *Bruggisser et al., 2017*; *Dimmer et al., 2012*; *Hulett et al., 2008*; *Käser et al., 2016*; *Lueder and Lithgow, 2009*; *Papic et al., 2011*; *Thornton et al., 2010*; *Waizenegger et al., 2005*). Whether the two oligomers directly catalyse protein insertion or whether they form microdomains in the MOM that facilitate membrane integration of helical segments is unclear. In any case, we hypothesise that the MIM complex and pATOM36 should behave similarly in this respect.

The successful complementation of the functions of the yeast MIM complex by trypanosomal pATOM36 and vice versa opens the way for future comparative studies to define the fundamental features the two biogenesis complexes share. The constraints imposed by their identical functions will help to reveal their mechanism of action. Taken together, our work offers new insights into the evolution of mitochondrial import factors and sheds new light on basic aspects of the biogenesis of mitochondrial outer membrane proteins.

# Materials and methods

## Key resources table

| Reagent type (species) or resource | Designation | Source or reference | Identifiers | Additional information |
|---|---|---|---|---|
| Strain, strain background (*Saccharomyces cerevisiae*) | WT; W303α; MATα leu2-3,112 trp1-1 can1-100 ura3-1 ade2-1 his3-11,15 | NA | | |
| Strain, strain background (*S. cerevisiae*) | *mim1Δ*; W303α; MATα leu2-3,112 trp1-1 can1-100 ura3-1 ade2-1 his3-11,15 MIM1::KanMX | DOI: 10.1242/jcs.103804 | | |
| Strain, strain background (*S. cerevisiae*) | *mim2Δ*; W303α; MATα leu2-3, 112 trp1-1 can1-100 ura3-1 ade2-1 his3-11,15 MIM2::HIS3 | DOI: 10.1242/jcs.103804 | | |
| Strain, strain background (*S. cerevisiae*) | *mim1Δ mim2Δ*; W303α; MATα leu2-3,112 trp1-1 can1-100 ura3-1 ade2-1 his3-11,15 MIM1::KanMX MIM2::HIS3 | DOI: 10.1242/jcs.103804 | | |
| Strain, strain background (*S. cerevisiae*) | WT; YPH499; MATa ura3-52 lys2-801_amber ade2-101 _ochre trp1-Δ63 his3-Δ200 leu2-Δ1 | | | |

*Continued on next page*

*Continued*

| Reagent type (species) or resource | Designation | Source or reference | Identifiers | Additional information |
|---|---|---|---|---|
| Strain, strain background (*S. cerevisiae*) | *mas37Δ*; YPH499; MATa ura3-52 lys2-801_amber ade2-101_ochre trp1-Δ63 his3-Δ200 leu2-Δ1 MAS37::HIS3 | DOI: 10.1074/jbc.M411510200 | | |
| Cell line (*Trypanosoma brucei*) | 29–13, procyclic, pATOM36 RNAi | PMID: 22787278 | | |
| Transfected construct (*S. cerevisiae*) | pATOM36 RNAi + mim1-myc/ mim2-HA (*Figures 6*, *7* and *8*) | this paper | | see Materials and methods |
| Transfected constructs (*S. cerevisiae*) | pATOM36 RNAi + mim1-myc/ mim2-HA (*Figure 8—figure supplement 1*) | this paper | | see Materials and methods |
| Antibody | anti-HA (polyclonal rat) | Roche | 11867423001; AB_390918 | WB 1:15000 |
| Antibody | anti-Tom70 (polyclonal rabbit) | N/A | | WB 1:2000 |
| Antibody | anti-Hep1 (polyclonal rabbit) | N/A | | WB 1:3000 |
| Antibody | anti-Ugo1 (polyclonal rabbit) | N/A | | WB 1:500 |
| Antibody | anti-Tom20 (polyclonal rabbit) | N/A | | WB 1:1600 |
| Antibody | anti-Fis1 (polyclonal rabbit) | N/A | | WB 1:1000 |
| Antibody | anti-Hsp60 (polyclonal rabbit) | N/A | | WB 1:100000 |
| Antibody | anti-Tom40 (polyclonal rabbit) | N/A | | WB 1:4000 |
| Antibody | anti-Aco1 (polyclonal rabbit) | N/A | | WB 1:7000 |
| Antibody | anti-Tom22 (polyclonal rabbit) | N/A | | WB 1:2000 |
| Antibody | anti-Tob55 (polyclonal rabbit) | N/A | | WB 1:2000 |
| Antibody | anti-Por1 (polyclonal rabbit) | N/A | | WB 1:4000 |
| Antibody | anti-rat (HRP coupled goat) | Abcam | ab6845; AB_955449 | WB 1:3000 |
| Antibody | anti-rabbit (HRP coupled goat) | Bio-Rad | 1721019; AB_11125143 | WB 1:10000 |
| Antibody | anti-myc (monoclonal mouse) | Invitrogen | 132500 | WB 1:2000 |
| Antibody | anti-HA (monoclonal mouse) | Enzo Life Sciences AG | CO-MMS-101 R-1000 | WB 1:5000 |
| Antibody | anti-EF1a (monoclonal mouse) | Merck Millipore | 05–235 | WB 1:10000 |
| Antibody | anti-ATOM40 (polyclonal rabbit) | N/A | | WB 1:10000, IF 1:1000 |
| Antibody | anti-CytC (polyclonal rabbit) | N/A | | WB 1:1000 |
| Antibody | anti-ATOM69 (polyclonal rabbit, affinity purified) | N/A | | WB 1:50 |
| Antibody | anti-TbTim9 (polyclonal rabbit) | N/A | | WB 1:20 |

*Continued on next page*

*Continued*

| Reagent type (species) or resource | Designation | Source or reference | Identifiers | Additional information |
|---|---|---|---|---|
| Antibody | anti-mtHsp70 (mouse) | N/A | | WB 1:1000 |
| Antibody | anti-ATOM46 (polyclonal rabbit; affinity purified) | N/A | | WB 1:50 |
| Antibody | anti-ATOM19 (mouse) | N/A | | WB 1:500 |
| Antibody | anti-ATOM14 (polyclonal rabbit) | N/A | | WB 1:500 |
| Antibody | anti-pATOM36 (polyclonal rabbit; affintiy purified) | N/A | | WB 1:250 |
| Antibody | anti-rabbit Alexa488 | ThermoFisher Scientific | | IF 1:1000 |
| Antibody | anti-rabbit IRDye 800CW | LI-COR Biosciences | P/N 925–32211 | WB 1:20000 |
| Antibody | anti-mouse IRDye LT680 | LI-COR Biosciences | P/N 925–68020; AB_2687826 | WB 1:20000 |
| Antibody | anti-mouse (HRP-coupled goat) | Sigma Aldrich | AP308P | WB 1:5000 |
| Antibody | anti-rabbit (HRP coupled goat) | Sigma Aldrich | AP307P | WB 1:5000 |
| Recombinant DNA reagent | Ø; pYX142 (plasmid) | | | |
| Recombinant DNA reagent | pATOM36; pYX142-pATOM36 (plasmid) | this paper | | pATOM36 ORF was amplified from pFT33 and cloned in pYX142 between EcoRI and BamHI |
| Recombinant DNA reagent | pATOM36-HA; pYX142-pATOM36-3HA (plasmid) | this paper | | pATOM36 ORF was amplified from pFT33 and cloned in pYX142 between EcoRI and BamHI |
| Recombinant DNA reagent | $^{35}$S-Mim1; pGEM4-Mim1-4M (plasmid) | DOI: 10.1038/sj.embor.7400318 | | |
| Recombinant DNA reagent | $^{35}$S-pATOM36; pGEM4-pATOM36 (plasmid) | this paper | | pATOM36 ORF was subcloned from pYX142-pATOM36 in pGEM4 with EcoRI and BamHI |
| Recombinant DNA reagent | $^{35}$S-Tom20$_{ext}$; pGEM4-Tom20$_{ext}$ (plasmid) | DOI: 10.1074/jbc.M410905200 | | |
| Recombinant DNA reagent | $^{35}$S-Tom20; pGEM3-Tom20 (plasmid) | DOI: 10.1074/jbc.M410905200 | | |
| Recombinant DNA reagent | $^{35}$S-Ugo1; pGEM4-Ugo1 (plasmid) | DOI: 10.1083/jcb.201102041 | | |
| Recombinant DNA reagent | $^{35}$S-Fis1; pGEM4-Fis1-TMC (plasmid) | DOI: 10.1242/jcs.024034 | | |
| Recombinant DNA reagent | $^{35}$S-pSu9-DHFR; pGEM4-pSu9-DHFR (plasmid) | PMID: 2892669 | | |
| Recombinant DNA reagent | mito-GFP; pRS426-TPI-pSu9-eGFP (plasmid) | this paper | | pSu9-eGFP was subcloned from pYX142-pSu9-GFP (Westermann B. and Neupert W. *Yeast*, 2000) to pRS426 with EcoRI and HindIII |
| Peptide, recombinant protein GST | GST | DOI: 10.1128/MCB.00227–13 | | |
| Peptide, recombinant protein GST-Tom70 | GST-Tom70 | DOI: 10.1128/MCB.00227–13 | | |

## Yeast strains and growth conditions

Yeast strains used in the study were isogenic to *Saccharomyces cerevisiae* strain W303α beside *mas37Δ*, which is isogenic to YPH499. Standard genetic techniques were used for growth and manipulation of yeast strains. Yeast cells were grown in synthetic medium S (0.67% [w/v] bacto-yeast nitrogen base without amino acids) with glucose (2% [w/v]), glycerol (3% [w/v]), or lactate (2% [w/v]) as carbon source. Transformation of yeast cells was performed by the lithium acetate method. Strains deleted for *MIM1*, *MIM2* or both were previously described (*Dimmer et al., 2012*). For drop-dilution assay, cells were grown in a synthetic medium to an $OD_{600}$ of 1.0 and diluted in fivefold increments followed by spotting 5 µl of the diluted cells on solid media.

## Transgenic cell lines and growth of *T. brucei*

Transgenic procyclic cell lines are based on *T. brucei* 29–13 cells (*Wirtz et al., 1999*) and were grown at 27°C in SDM-79 medium supplemented with 10% FCS (v/v). The RNAi cell line targeting the open reading frame of pATOM36 (Tb927.7.5700, Q582I5) was previously described (*Pusnik et al., 2012*). For growth curves, tetracycline induced and uninduced cell lines were diluted to $2 \times 10^6$ cells/ml every 2 days and the cumulative cell number was calculated.

## Recombinant DNA techniques

pATOM36 and its 3xHA-tagged variant were cloned into the yeast expression plasmid pYX142-TPI-$_{pro}$ using the EcoRI and BamHI cutting sites. For simultaneous and inducible expression of *S.c.* Mim1-myc and Mim2-HA in *T. brucei,* the appropriate cell line was transfected with a pLew100-based plasmid (*Bochud-Allemann and Schneider, 2002*; *Wirtz et al., 1999*). For optimal expression of the proteins, the ORFs were adapted to the codon usage of *T. brucei* according to *Horn (2008)*. The intergenic region of the α- and β-tubulin genes was cloned in between the ORFs. The insert was synthesised by GenScript with flanking HindIII and BamHI sites for cloning into the pLew100 vector.

For expression from distinct plasmids, the ORFs of *MIM1* and *MIM2* were amplified from yeast genomic DNA and cloned into pLew100-based expression vectors using HindIII and BamHI for *MIM1* and HindIII and XbaI for *MIM2*.

## Biochemical methods

Protein samples for immunodecoration were analysed on 8, 12, 12.5, or 15% SDS-PAGE and subsequently transferred onto nitrocellulose membranes by semi-dry western blotting. Proteins were detected by incubating the membranes first with primary antibodies and then with either horseradish peroxidase-conjugates of goat anti-rabbit, goat anti-mouse or goat anti-rat secondary antibodies or with secondary antibodies coupled to fluorescent dye and usage of the LI-COR system.

Isolation of mitochondria from yeast cells was performed by differential centrifugation, as previously described (*Daum et al., 1982*). For protease protection assay, 50 µg of mitochondria were resuspended in 100 µl of SEM buffer (250 mM sucrose, 1 mM EDTA, 10 mM MOPS, pH 7.2). As a control, mitochondria were treated with 1% Triton X-100 in SEM buffer and incubated on ice for 30 min. The samples were supplemented with Proteinase K (50 µg/ml) and incubated on ice for 30 min. The proteolytic reaction was stopped with 5 mM Phenylmethylsulfonyl fluoride (PMSF). The samples were precipitated with trichloroacetic acid (TCA) and resuspended in 40 µl of 2x Laemmli buffer, heated for 10 min at 95°C, and analysed by SDS-PAGE and immunoblotting.

To analyse the membrane topology of proteins, alkaline extraction was performed. Mitochondria (50 µg) were resuspended in 100 µl of buffer containing 10 mM HEPES-KOH, 100 mM $Na_2CO_3$, pH 11.5 and incubated 30 min on ice. The membrane fraction was pelleted by centrifugation (76000xg, 30 min, 2°C) and the supernatant fraction was precipitated with TCA. Both fractions were resuspended in 40 µl of 2x Laemmli buffer, heated for 10 min at 95°C, and analysed by SDS-PAGE and immunoblotting.

GST-pulldown with radiolabelled proteins was performed as previously described (*Papić et al., 2013*).

For mitochondria enriched fractions by digitonin extraction of *T. brucei*, the cells were incubated for 10 min on ice in 20 mM Tris-HCl pH 7.5, 0.6 M sorbitol, 2 mM EDTA containing 0.025% (w/v) digitonin. After centrifugation (6,800 g, 4°C), the resulting mitochondria enriched fraction was separated

from the supernatant and subjected to SDS-PAGE. The mitochondria enriched pellets were also used for further experiments.

### In vitro synthesis and mitochondrial import of radiolabelled proteins

In vitro transcription was performed with SP6 polymerase from either pGEM4 or pGEM3 plasmid encoding the gene of interest. Proteins were then in vitro translated from the acquired mRNA in the presence of $^{35}$S-methionine in rabbit reticulocyte lysate (Promega, Madison, WI, USA). Protein import was performed by adding 50 µg of isolated organelles to 100 µl of import buffer harboring 1 mM NADH and 2 mM ATP. Then, the translation reaction was added to the mitochondria solution and import of precursor proteins was performed at either 25°C for pSu9-DHFR, Tom20 and Ugo1 or at 2°C for Fis1 and Tom20$_{ext}$. Import of Tom20, Fis1-TMC, and Ugo1 was monitored according to established assays (*Ahting et al., 2005*; *Kemper et al., 2008*; *Papic et al., 2011*).

### Blue native gel electrophoresis (BN-PAGE)

Assembly of native complexes was analysed by BN-PAGE. Mitochondria or mitochondria-enriched fractions were solubilised with buffer (1% digitonin or 0.2% TritonX-100, 20 mM Tris, 0.1 mM EDTA, 50 mM NaCl, 10% glycerol, pH 7.4) for 30 min at 4°C on an overhead shaker. After a clarifying spin (30,000xg, 15 min, 2°C), 10x sample buffer (5% [wt/vol] Coomassie brilliant blue G-250, 100 mM Bis-Tris, 500 mM 6-aminocaproic acid, pH 7.0) was added and the mixture was analysed by electrophoresis in a blue native gel containing either 6–14% or 8–13% gradient of acrylamide (*Schägger et al., 1994*). To analyse the assembly of radiolabelled Tom20 molecules, the organelles were solubilised with 0.2% digitonin. BN-PAGE was followed by either western blotting or autoradiography. The mixture NativeMark Unstained Protein Standard was used to monitor the migration of molecular weight marker proteins.

### Fluorescence microscopy

Fluorescence images of yeast cells were acquired with spinning disk microscope Zeiss Axio Examiner Z1 equipped with a CSU-X1 real-time confocal system (Visitron, Puchheim, Germany), VS-Laser system, and SPOT Flex CCD camera (Visitron Systems). Images were analysed with VisiView software (Visitron). Immunofluorescence images of *T. brucei* were acquired with a DFC360 FX monochrome camera (Leica Microsystrems, Nussloch, Germany) and a DMI6000B microscope (Leica Microsystems). Image analysis was done using LAS X software (Leica Microsystems), ImageJ, and Adobe Photoshop CS5.1 (Adobe).

## Acknowledgements

We thank E Kracker for technical assistance. This work was supported by the Deutsche Forschungsgemeinschaft (RA 1028/7–1,2 and DIP to DR), the ITN TAMPting to DV and DR (funded by the Marie Curie Actions of the EU [grant number 607072]). KSD was supported by the PROFILplus program of the Faculty of Medicine of the University of Tübingen. Research in the Schneider group was supported by grant 175563 and in part by the NCCR 'RNA and Disease' both funded by the Swiss National Science Foundation.

## Additional information

### Funding

| Funder | Grant reference number | Author |
| --- | --- | --- |
| H2020 Marie Skłodowska-Curie Actions | ITN TAMPting, 607072 | Daniela G Vitali<br>Doron Rapaport |
| Schweizerischer Nationalfonds zur Förderung der Wissenschaftlichen Forschung | 138355 | Andre Schneider |
| Deutsche Forschungsgemeinschaft | RA 1028/7-1,RA 1028/10-1 | Doron Rapaport |

The funders had no role in study design, data collection and interpretation, or the decision to submit the work for publication.

### Author contributions
Daniela G Vitali, Conceptualization, Investigation, Methodology; Sandro Käser, Conceptualization, Investigation, Methodology, Writing—original draft; Antonia Kolb, Investigation, Methodology; Kai S Dimmer, Resources, Funding acquisition; Andre Schneider, Doron Rapaport, Conceptualization, Supervision, Funding acquisition, Writing—original draft

### Author ORCIDs
Doron Rapaport http://orcid.org/0000-0003-3136-1207

### Decision letter and Author response
Decision letter https://doi.org/10.7554/eLife.34488.018
Author response https://doi.org/10.7554/eLife.34488.019

## Additional files

### Data availability
All data generated or analysed during this study are included in the manuscript and supporting files.

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
