## [Decision Letter]

Thank you for submitting your article "Independent evolution of functionally exchangeable mitochondrial outer membrane import complexes" for consideration by *eLife*. Your article has been favorably evaluated by Vivek Malhotra (Senior Editor) and three reviewers, one of whom, Nikolaus Pfanner, is a member of our Board of Reviewing Editors. The following individual involved in review of your submission has agreed to reveal his identity: Kostas Tokatlidis (Reviewer #2).

The reviewers have discussed the reviews with one another and the Reviewing Editor has drafted this decision to help you prepare a revised submission.

Summary:

Mitochondria are iconic structures of eukarytotic cells and are involved in many important functions including oxidative phosphorylation. The greater part of the mitochondrial proteome derives from nuclear-encoded proteins, which are synthesized in the cytosol and subsequently imported into mitochondria. The MIM pathway (consisting of the two outer membrane proteins Mim1/Mim2) is a fungal specific-pathway required for the biogenesis of a subset of outer membrane proteins, including subunits of the general entry gate of the organelle the TOM complex. pATOM36 is a recently identified trypanosomatid-specific essential mitochondrial outer membrane protein that functions in the biogenesis of outer membrane proteins and transmission of mitochondrial DNA. In this study Vitali and colleagues show that Mim1/Mim2 and pATOM36 are functionally interchangeable by using complementation analysis in yeast and *T. brucei* together with functional analyses of import defects.

The study is important as no functional homologs of the fungal Mim1/2 system have yet been identified in any other eukaryotes. Remarkably, pATOM36 and Mim1/2 share no sequence homology and are different in their topology and possibly structure. The results reveal that MIM and pATOM36 are the products of convergent evolution, which arose after the ancestors of fungi and trypanosomes diverged.

The work presented in this manuscript is thorough and explores a crucial question of mitochondrial biology, representing the first case of convergent evolution for the mitochondrial protein import machinery.

Essential revisions:

1) The authors present a number of data indicating that pATOM36 may compensate for the loss of the MIM machinery but do not directly show a role of pATOM36 in protein import into the outer membrane of yeast mitochondria. The authors should perform in vitro import assays into isolated yeast mitochondria to show that the import of MIM substrates, including specific TOM subunits, is restored upon expression of pATOM36 in MIM mutant strains. Are MIM-independent protein import pathways impaired in the MIM double mutant and are they affected by expression of pATOM36?

Figure 3A: The levels of Ugo1 and Tom70 are not completely restored upon expression of pATOM36, when compared to the levels of Tom20. A quantification of these Western blots will be helpful. It would be valuable to see how the in vitro import and assembly of these substrates is influenced in these cells, as it might reveal that pATOM36 might have a preference for certain MIM substrates.

2) Given that the protein biogenesis function of pATOM36 is substantially restored one wonders whether only MIm1 could do this in trypanosomes or whether there is an additive effect of MIm1 and Mim2 in these studies.

Why does expression of Mim1/Mim2 not complement the loss of kDNA? Does this have something to do with the considerably larger size/topology of pATOM36? Does this imply that only the TMDs of pATOM36 are involved in the biogenesis of outer membrane proteins?

3) Does pATOM36 expressed in yeast cooperate with Tom70, as has been shown for Mim1?

4) Many of the BN blots (for instance Figure 3C) are saturated and need be shown as lower-exposures.

---

## [Author Response]

Essential revisions:1) The authors present a number of data indicating that pATOM36 may compensate for the loss of the MIM machinery but do not directly show a role of pATOM36 in protein import into the outer membrane of yeast mitochondria. The authors should perform in vitro import assays into isolated yeast mitochondria to show that the import of MIM substrates, including specific TOM subunits, is restored upon expression of pATOM36 in MIM mutant strains. Are MIM-independent protein import pathways impaired in the MIM double mutant and are they affected by expression of pATOM36?

As was proposed, we performed in vitro import assays of radiolabeled MIM substrates like Ugo1 and Tom20. We did not test Tom70 import as we are not aware of a specific and reliable in vitro assay to monitor the correct membrane integration of this protein. In agreement with the changes in the steady state levels (see below), the expression of pATOM36 did not result in a significant effect on the extent of import of Ugo1 but improved significantly the import of Tom20 in the mim1Δ/mim2Δ double deletion strain. The presence of pATOM36 did not affect the import of control non MIM substrates like pSu9-DHFR or Fis1. These new findings are included in revised Figure 4 and are described and discussed in the revised Results and Discussion sections.

Figure 3A: The levels of Ugo1 and Tom70 are not completely restored upon expression of pATOM36, when compared to the levels of Tom20. A quantification of these Western blots will be helpful. It would be valuable to see how the in vitro import and assembly of these substrates is influenced in these cells, as it might reveal that pATOM36 might have a preference for certain MIM substrates.

We followed the suggestion of the reviewers and quantified the immunodecorations of the MIM substrates. Indeed, the effect of pATOM36 on such substrates is variable (revised Figure 3B). Whereas, the expression of pATOM36 in the mim1Δ/mim2Δ double deletion strain elevated significantly the levels of Tom70 and especially of Tom20, those of Ugo1 were only mildly increased. Hence, considering also the in vitro import assays (see above), it seems that pATOM36 has a preference for certain MIM substrates like Tom20. These new findings are described and discussed in the revised Results and Discussion sections.

2) Given that the protein biogenesis function of pATOM36 is substantially restored one wonders whether only MIm1 could do this in trypanosomes or whether there is an additive effect of MIm1 and Mim2 in these studies.

We could identify clones of pATOM36-RNAi cell lines, which unintentionally express only tagged Mim1 or Mim2 in major amounts and in which the other partner protein was barely detected. In such clones, we could neither detect rescue of growth retardation nor restoration of the ATOM assembly defect. Thus, we conclude that restoration of the ATOM complex biogenesis defect requires both MIM subunits. These new results are shown in the newly added supplementary Figure 8—figure supplement 1 of the revised manuscript and are discussed at the end of the Results section (new section entitled "Complementing the biogenesis function of pATOM36 requires Mim1 and Mim2"). In addition, relevant information was added to the Materials and methods” section.

Why does expression of Mim1/Mim2 not complement the loss of kDNA? Does this have something to do with the considerably larger size/topology of pATOM36?

It has previously been shown that pATOM36 is present in three distinct complexes of approximately 140, 250 and 480 kDa (Käser et al., 2016). Based on the present study, the 140 and 250 kDa complexes likely represent homo-oligomers of pATOM36. Co-immunoprecipitations of tagged pATOM36 with *T. brucei* cells identified a number of interacting proteins of the 480 kDa complex. Many of these are subunits of the tripartite attachment complex (TAC), which links the kDNA to the basal body of the flagellum and is essential for kDNA maintenance. The TAC is a multiprotein complex of which presently seven subunits were characterized. Each of these subunits is specific for trypanosomatids and essential for TAC function. Thus, pATOM36 needs to interact with other TAC subunits in order to function in the TAC, whereas it does not require other proteins to perform its role in outer membrane protein biogenesis. Our findings that expression of Mim1 and Mim2 selectively complements the outer membrane protein biogenesis but not the mitochondrial DNA inheritance phenotype is therefore expected. It would indeed be very surprising if Mim1 and Mim2 would interact with trypanosomatid-specific TAC subunits.

Does this imply that only the TMDs of pATOM36 are involved in the biogenesis of outer membrane proteins?

This is a complex question since, as stated in the manuscript, we do not know how many transmembrane domains pATOM36 really has. However, it is known that pATOM36 contains domains specialized for only one of its two functions. Previous complementation experiments have shown that the C-terminal 75 amino acids, which contain the transmembrane domain, which is predicted with the highest score, are dispensable for mitochondrial DNA inheritance but are required for the biogenesis of the ATOM complex (Käser et al., 2016).

In order to test further this point, we aimed to expressed in yeast cells the pATOM36 variant that lacks the 75 C-terminally amino acids. Unfortunately, this variant was detected in very low levels, suggesting that it is unstable in the yeast environment. Therefore, its inability to complement the mim1Δ/mim2Δ growth defect cannot be interpreted.

3) Does pATOM36 expressed in yeast cooperate with Tom70, as has been shown for Mim1?

We performed several experiments in order to address this point. Initially, we solubilized mitochondria from yeast cells expressing pATOM36-HA and tested whether Tom70 can be pulled-down by the former protein. We could not observe any specific binding, probably because the interaction under physiological conditions is transient. It should be noted that His tagged-Tom70 could pull-down only minor amounts of Mim1 (<0.5% of total Mim1) in digitonin-solubilized mitochondria (Becker et al., 2011).

Next, to test the potential of the two proteins to interact with each other, we used recombinant GST-Tom70 cytosolic domain (and GST alone as control) and incubated them with in vitro synthesized radiolabeled Mim1 or pATOM36. We detected specific interaction of Tom70 with both proteins. Thus, although this outcome is not an evidence for an in vivo interaction, it demonstrates that pATOM36 and Tom70 can interact with each other. Naturally, we cannot exclude the possibility that Tom70 recognizes either Mim1 or pATOM36 as substrates rather than as interaction partners. These new results are included in revised Figure 3E and are described in the revised Results sections.

4) Many of the BN blots (for instance Figure 3C) are saturated and need be shown as lower-exposures.

We corrected this point and the BN-PAGE immunodecorations in revised Figures 1B, 3D, and Figure 3—figure supplement 1A are shown with lower exposures.